# Psoas Muscle Index Predicts Perioperative Mortality in Patients Undergoing LVAD Implantation

**DOI:** 10.3390/muscles4030041

**Published:** 2025-09-22

**Authors:** Steven Hopkins, Paulomi Gohel, Sakshi Gandhi, Moiz Nasir, William Stallings, Eman Hamad

**Affiliations:** 1Department of Internal Medicine, University of Pittsburgh Medical Center, Pittsburgh, PA 15213, USA; hopkinss6@upmc.edu (S.H.); gohelp2@upmc.edu (P.G.); 2Division of Cardiology, University of Maryland Medical Center, Baltimore, MD 21201, USA; 3Department of Internal Medicine, University of Virginia Medical Center, Charlottesville, VA 22908, USA; moiznasir@gmail.com; 4Heart and Vascular Institute, Temple University Hospital, Philadelphia, PA 19140, USA

**Keywords:** sarcopenia, psoas muscle index, LVAD, heart failure

## Abstract

Background: Sarcopenia is associated with adverse surgical outcomes across multiple specialties. The psoas muscle index (PMI), a radiologic marker of sarcopenia, may offer prognostic value in patients undergoing left ventricular assist device (LVAD) implantation, a population frequently characterized by frailty and high perioperative risk. Methods: We conducted a single-center retrospective study of 32 patients who underwent LVAD implantation between 2017 and 2022 and had preoperative CT imaging within 45 days. PMI was calculated from bilateral psoas muscle area at the L3 vertebral level, normalized to height. Sarcopenia was defined as the lowest sex-specific quartile of PMI. Primary outcomes were overall survival (OS), 90-day mortality, and postoperative length of stay (LOS). Results: Eight patients (25%) met criteria for sarcopenia. Sarcopenic and non-sarcopenic groups had similar demographics, comorbidities, and nutritional status. While there were no significant differences in overall, 90-day, or 1-year mortality between groups, among those who died post-implantation, the sarcopenic group had significantly shorter OS (median 38 vs. 597 days, *p* = 0.006). All sarcopenic deaths occurred within 90 days post-implant. LOS did not differ significantly between groups. Conclusions: PMI-defined sarcopenia was associated with early postoperative mortality among LVAD recipients, though not with overall or long-term mortality.

## 1. Introduction

Heart failure (HF), affecting 64 million people worldwide, continues to be one of the leading causes of global morbidity, mortality, and elevated healthcare cost burden [1]. Advanced HF not only compromises cardiovascular performance but also exerts profound effects on skeletal muscle, metabolism, and nutritional status, thereby contributing to a vicious cycle of functional decline and frailty [2]. Cardiac cachexia, a multifactorial syndrome comprising muscle wasting and poor nutritional status, has long been recognized as a manifestation of end-stage HF [2]. Despite advancements in medical therapy, the standard of care of patients with end-stage HF remains orthotopic heart transplant (OHT), with post-transplant survival rates roughly estimated at 85% [3]. Many transplant centers now consider frailty in the setting of cardiac cachexia as a critical factor in the evaluation of heart transplant candidacy, often disqualifying patients who demonstrate significant frailty from receiving OHT [3,4]. Frailty assessment has thus evolved into a key determinant in advanced HF management pathways, influencing both candidacy decisions and perioperative planning.

Left ventricular assist devices (LVADs) have emerged as an alternative mechanical circulatory support intervention for patients who may not be optimal OHT candidates [5]. Initially intended as a bridge to transplantation (BTT), LVADs are increasingly being utilized as destination therapy (DT), with nearly half of all devices now implanted intended for DT [6]. In 2018, the International Registry for Mechanically Assisted Circulatory Support (IMACS) reported survival of DT LVAD patients to be 80% and 70% at one and two years, respectively [7].

Despite advancements in LVAD technology and improved survival rates, implantation remains associated with significant risks, given the frailty and co-morbidities frequently seen in end-stage HF populations [8]. Moreover, the ability to predict survival following LVAD implantation remains limited, as prognostic tools have yet to be fully developed or validated [9]. Identifying factors predictive of poor outcomes in this vulnerable patient population is essential for optimizing pre- and perioperative management and improving candidacy selection criteria. In this regard, objective measures of musculoskeletal health, particularly those that can be derived from standard-of-care imaging, hold promise as scalable, reproducible, and clinically relevant risk models.

Sarcopenia, defined as the progressive loss of skeletal muscle mass, strength, and functional capacity, has been consistently linked to adverse surgical postoperative outcomes and increased morbidity [10]. Multiple modalities exist for the assessment of sarcopenia, including functional tests such as hand grip strength (HGS) and nutritional evaluation tools like the Mini Nutritional Assessment (MNA). However, radiological imaging has continued to be a preferred method due to its capacity to provide accurate, objective, and reproducible measurements of muscle mass through direct visualization of body composition [11]. An additional advantage is that such imaging is often performed for other clinical indications, allowing for retrospective sarcopenia assessment without additional cost or patient burden [11].

In recent years, the psoas major muscle has been utilized as a metric of sarcopenia. The psoas plays a critical role in maintaining posture and supporting the spine and pelvis and is readily and reliably identified on cross-sectional imaging, especially computed tomography (CT), making it an important metric point for sarcopenia [12]. The Psoas Muscle Index (PMI), calculated as the cross-sectional area of both psoas muscles at the third lumbar vertebral (L3) level normalized to patient height, has been shown to correlate strongly with total lean skeletal muscle mass. PMI has demonstrated prognostic relevance across various cardiovascular diseases [13] and in patients undergoing both open surgical interventions and minimally invasive procedures [14]. Its anatomic consistency, resistance to measurement variability, and strong correlation with whole-body muscle mass make it an especially attractive biomarker in patients with chronic illnesses, as seen in patients with chronic HF and sarcopenia.

Despite these findings, the prognostic value of PMI in patients undergoing LVAD and OHT implantation remains unexplored. This study seeks to evaluate the prognostic significance of the Psoas Muscle Index (PMI) in predicting overall survival (OS) and postoperative length of stay (LOS) among patients undergoing LVAD evaluation. By focusing on LVAD recipients, a population characterized by significant physiologic stress, surgical complexity, and heterogeneous recovery potential, we aim to clarify whether PMI-diagnosed sarcopenia can contribute to risk stratification in overall survival and chronological mortality risk. We hypothesize that sarcopenic status, as defined by PMI, would be predictive of poorer outcomes, including lower OS and prolonged LOS. By investigating the role of PMI as predictive of clinical outcomes, this study seeks to inform clinical practice by identifying patients who may benefit from preoperative nutritional intervention and to contribute to the growing body of evidence on the importance of preoperative risk stratification in LVAD candidates. If validated, PMI assessment could be seamlessly incorporated into preoperative workflows, tailoring prehabilitation regimens prior to device implantation.

## 2. Methods

This study was a single-center retrospective analysis. The study population consisted of 41 patients who underwent LVAD implantation between 2017 and 2022. As this was a retrospective study, the sample size was determined by the number of patients who underwent LVAD implantation with preoperative CT imaging available within 45 days. No a priori power calculation was performed, and our analysis is therefore exploratory in nature. Psoas muscle area was measured at the L3 level of the CT scan, and PMI was then calculated using the formula (left psoas area + right psoas area)/height^2^ [15]. The choice of the L3 vertebral level is supported by prior studies demonstrating a strong correlation between muscle cross-sectional area at this site and total body skeletal muscle mass, as well as its generalizability across studies [3].

Inclusion criteria were patients who underwent LVAD implantation between 2017 and 2022 and had a preoperative CT abdomen/pelvis within 45 days of surgery. Exclusion criteria were patients without available CT imaging in this timeframe or without at least one year of follow-up data. These criteria were chosen to balance the proximity of imaging to the operative event with feasibility in clinical practice. Exclusion criteria included (1) patients without computed tomography abdomen/pelvis (CTAP) imaging within 45 days of implantation and (2) patients without one year of follow-up information after implantation. The 45-day imaging window was selected to balance the need for proximity to the surgical event while allowing flexibility for clinical scheduling constraints. This interval has also been adopted in prior sarcopenia studies as a reasonable compromise between precision and feasibility [3]. Our study was approved by our Internal Review Board (IRB) (IRB protocol number 30546) and Research Review Committee (RRC).

Study data were collected and managed using REDCap (Research Electronic Data Capture) electronic data capture tools hosted at Temple University Hospital. We collected information regarding patient demographics, comorbidities, social history, date of diagnosis, hand grip strength (HGS), mini-nutritional assessment (MNA), and outpatient milrinone use. The principal outcome measures gathered were postoperative LOS, overall mortality, deceased within 90 days of implantation, deceased within 1 year of implantation, and deceased with the same admission as implantation.

### 2.1. Definition of Sarcopenia

Pre-interventional CT scans were accessed using eRAD PACS Viewer Version 7. PMI was obtained by measuring the cross-sectional area of bilateral psoas muscles at the level of L3 (Figure 1). PMI was calculated as the psoas muscle area divided by the squared patient height in units of cm^2^/m^2^. Presently, there is no consensus regarding optimal PMI cutoff values for sarcopenia, which vary in different populations [16]. In this study, sarcopenia was defined as the lowest sex-specific quartile PMI at pre-interventional CT scan, in accordance with what was performed in previous studies [16,17,18,19].

1st quartile PMI values were calculated as 4.99 cm^2^/m^2^ for males and 4.09 cm^2^/m^2^ for females, respectively. Patients who fell in the 1st quartile PMI for their gender were classified as sarcopenic for comparison with non-sarcopenic patients.

### 2.2. Statistical Analysis

The characteristics of the sarcopenic and non-sarcopenic OHT and LVAD recipients were compared using *T*-tests and the chi-squared test. A prespecified 2-sided alpha of 0.05 and 95% confidence intervals (CIs) were used to determine statistical significance. All statistical analyses were performed using IBM SPSS Statistics Version 29.0.0.0 (IBM Corp. Released 2022. IBM SPSS Statistics for Windows, Version 29.0. Armonk, NY, USA: IBM Corp.).

## 3. Results

### 3.1. Cohort and Sarcopenia Distribution

A total of 108 cases of LVAD recipients were reviewed, of which we identified 32 patients with computed tomography of the abdomen and pelvis (CTAP) within 45 days of implantation. Sarcopenia was present in eight (25%) of LVAD patients. The 32-patient population was divided into an 8-patient sarcopenic cohort and a 24-patient non-sarcopenic cohort (Figure 2). The distribution of PMI values demonstrated a right-skewed pattern, with most patients clustering near the median and a smaller subset representing extreme sarcopenia, suggesting a spectrum of muscle depletion within the cohort.

### 3.2. Baseline Demographics and Comorbidities

Cohort demographics and comorbidities are displayed in Table 1. There were no statistically significant differences between sarcopenic and non-sarcopenic cohorts regarding gender (87.5% male vs. 70.8%, *p* = 0.346), ethnicity (62.5% White vs. 29.2%, *p* = 0.396), or age at transplant (60 years vs. 55.8, *p* = 0.271). Regarding comorbidities, there were no significant differences between the sarcopenic and non-sarcopenic cohorts regarding chronic kidney disease (75% vs. 75%, *p* = 1), atrial fibrillation (25% vs. 20.8%, *p* = 0.805), myocardial infarction (37.5% vs. 20.8%, *p* = 0.660), liver disease (37.5% vs. 20.8%, *p* = 0.660), cerebral vascular accident (37.5% vs. 45.8%, *p* = 0.533), or outpatient milrinone use (37.5% vs. 45.8%, *p* = 0.681). There was, additionally, no difference between cohorts regarding the proportion receiving LVAD as BTT (12.5% vs. 12.5%, *p* = 1).

### 3.3. Nutrition and Functional Assessments

Cohort nutritional and physical function markers are displayed in Table 2. The average BMI was slightly higher in the sarcopenic cohort than in the non-sarcopenic (30.6 vs. 29.6, *p* = 0.353). Albumin levels were marginally higher in the sarcopenic group (3.23 g/dL vs. 3.17, *p* = 0.403). Similarly, there was no significant difference in pre-albumin between cohorts (18.36 g/dL vs. 19.26, *p* = 0.389). Grip strength was lower in the sarcopenic group (28.23 kg) compared to the non-sarcopenic group (29.59 kg), with a *p*-value of 0.796. At-risk and malnourished MNA score status, defined as MNA scores of 17–23.5 and less than 17, respectively, was slightly higher in the sarcopenic group (62.5%) compared to the non-sarcopenic group (66.7%), with no significant difference (*p* = 0.830). Interestingly, despite lower muscle mass, sarcopenic patients did not universally exhibit lower serum nutritional markers, underscoring the fact that laboratory parameters may not fully capture functional muscle deficits in this population.

### 3.4. Clinical Outcomes

Cohort outcomes are displayed in Table 3. There were no differences between the sarcopenic cohort and non-sarcopenic cohort regarding overall mortality (37.5% vs. 37.5%, *p* = 1), 90-day mortality (37.5% vs. 8.3%, *p* = 0.160), or 1-year mortality (37.5% vs. 16.6%, *p* = 0.459). There were additionally no differences between the proportion of deceased same admission as LVAD implantation (12.5% vs. 8.3%, *p* = 0.726) or postoperative LOS (26.43 vs. 23.33, *p* = 0.623). OS (days) amongst deceased of both groups was 38 days for sarcopenic and 597 days for non-sarcopenic patients (*p* = 0.006). Kaplan–Meier survival data is displayed in Table 4, and the survival curve is displayed in Figure 3. Kaplan–Meier analysis demonstrated early divergence of survival curves within the first three postoperative months, followed by a plateau phase in which survival probabilities converged, consistent with the hypothesis that sarcopenia’s impact is most pronounced in the immediate perioperative period.

## 4. Discussion

### 4.1. Key Findings

This study demonstrates that while baseline demographics and comorbidities were similar between sarcopenic and non-sarcopenic LVAD recipients, there was a striking difference in early postoperative survival among those who died after implantation. Specifically, the sarcopenic cohort exhibited a median OS of just 38 days compared to nearly 600 days in the non-sarcopenic group. Although this difference did not translate into statistically significant disparities in long-term or overall mortality rates, the early postoperative hazard was markedly higher in the sarcopenic group. This pattern suggests that sarcopenia exerts its most profound influence during the high-stress perioperative window, a period characterized by surgical trauma, inflammatory activation, hemodynamic adjustments, and rapid metabolic shifts. Once this acute vulnerability period is survived, patients, regardless of preoperative sarcopenia status, appear to achieve a more stable survival trajectory under LVAD support.

### 4.2. Comparison with Existing Literature

In our study, PMI-defined sarcopenia was associated with significantly lower OS among LVAD recipients who died postoperatively, with all sarcopenic deaths occurring within 90 days of implantation. This reinforces the concept that skeletal muscle mass is not merely a reflection of nutritional status but a surrogate for the body’s ability to withstand and recover from major surgical stressors [4,5,6]. Our findings parallel those of prior investigations linking low muscle mass on imaging to worse short-term outcomes in cardiac surgery and other high-risk procedures [20,21,22,23]. However, by focusing on LVAD implantation, a procedure uniquely demanding in both surgical complexity and postoperative adaptation, we provide novel insight into sarcopenia’s prognostic role in this specific patient population.

Sarcopenia, as defined by skeletal muscle mass on imaging, has been consistently linked to worse perioperative outcomes. Recently, Yang et al. demonstrated sarcopenia as determined by skeletal muscle index (SMI) on CTAP to be associated with increased 30-day and 90-day postoperative mortality [20]. PMI has been shown to be correlated with greater LOS and increased need for intensive care and postoperative blood transfusion [21,22]. Batista et al. additionally have linked PMI-defined sarcopenia to increased 2-year mortality following major intra-abdominal colorectal surgery [23]. Additionally, Kumar et al. found that sarcopenia defined by low PMI was a strong predictor of 2-year mortality after LVAD implantation. Moreover, each unit increase in PMI was found to be significantly protective (HR ≈ 0.38, *p* = 0.001) [21]. Altogether, these studies underscore the prognostic value of PMI beyond the immediate postoperative period and across diverse populations and procedures. To date, however, our study is the first to investigate PMI as a prognosticator of perioperative mortality in LVAD recipients.

### 4.3. Potential Mechanisms

These results have direct relevance to LVAD candidate selection, preoperative optimization, and postoperative rehabilitation strategies. Skeletal muscle mass plays a vital role in metabolic and physical recovery following major surgery [24,25]. Muscle wasting has deleterious effects on the body’s ability to recover from the physiological stressors of surgery due in part to impaired mobilization of protein stores, resulting in prolonged inflammation, hemodynamic instability, and mechanical ventilation [26]. Moreover, heart failure–related systemic inflammation amplifies these issues, promoting a chronic catabolic state that impairs protein synthesis and further delays functional recovery [27].

Toth et al. demonstrated that in chronic heart failure, immobility and reduced activity levels contribute to a disproportionate loss of oxidative type I muscle fibers, mitochondrial dysfunction, and diminished capillary density, which together impair endurance capacity and further accelerate sarcopenia. This muscle phenotype shift reduces fatigue resistance and may limit postoperative rehabilitation potential even when cardiac output is restored by LVAD support [28]. Furthermore, this is compounded by HF-associated anabolic resistance, with blunted growth hormone (GH) and insulin-like growth factor-1 (IGF-1) signaling, which limits new protein synthesis [29]. These pathophysiologic changes can explain why sarcopenic LVAD recipients are disproportionately vulnerable in the early postoperative phase, which further emphasizes the importance of early targeted interventions to preserve and restore skeletal muscle during that time.

### 4.4. Impact on Rehabilitation and Recovery

Cardiac rehabilitation (CR) following LVAD implantation is a critical component of the recovery process. For non-sarcopenic patients, participation in CR has been associated with improved exercise tolerance, reduced hospital readmission rates, and better long-term survival [30]. These benefits stem from improved peripheral oxygen utilization, increased hemodynamic adaptations, and reversal of deconditioning inherent to advanced heart failure. However, sarcopenic patients, limited by reduced muscle mass, strength, and functional endurance, many times cannot engage fully in standard CR regimens, leading to suboptimal rehabilitation [31].

Shakuta et al. showed that sarcopenic patients who failed to improve during CR faced significantly higher all-cause mortality, suggesting that recovery of muscle mass may be a critical determinant of long-term prognosis. This results in a vicious cycle of inability to participate in CR, perpetuating muscle loss, increased frailty, and exacerbated vulnerability to postoperative complications [30]. For this reason, early identification of sarcopenia is critical, and the development of tailored interventions such as resistance training, nutritional optimization, and individualized rehabilitative programs may help LVAD recipients maximize both physiological recovery and overall survival [32]. This approach is supported by evidence from other surgical populations in which preoperative conditioning improved muscle mass and reduced postoperative complications [28,29,30].

### 4.5. Long-Term Implications

Although our study found no significant difference in long-term survival between sarcopenic and non-sarcopenic patients after 90 days, the early perioperative period appears to be a critical window in which sarcopenia exerts its greatest influence. This likely reflects the compounded impact of surgical stress, pre-existing frailty, and the metabolic demands of advanced HF, all of which disproportionately affect patients with diminished skeletal muscle reserves. Once sarcopenic patients survive this vulnerable phase, they may benefit from hemodynamic improvements provided by the LVAD. This hemodynamic stabilization, coupled with participation in structured cardiac rehabilitation, could allow sarcopenic patients to regain strength and functional capacity to equal that of their non-sarcopenic cohorts.

The current literature suggests that the improved cardiac output and end-organ perfusion afforded by LVAD support can attenuate the chronic catabolic state seen in advanced HF, leading to an adequate increase in lean muscle mass, reversal of oxidative fiber loss, and improved mitochondrial function when coupled with prehabilitation and perioperative optimization. In some cohorts, postoperative increases in muscle mass have been associated with better functional performance and reduced rehospitalization risk, underscoring that sarcopenia in the LVAD population may be a modifiable, not fixed, risk factor given appropriate interventions.

Thus, early identification of sarcopenia and preoperative interventions aimed at improving muscle mass and strength may be essential for optimizing participation in post-LVAD cardiac rehabilitation. PMI, given its ease of measurement and validated correlation to sarcopenic status, provides an accessible and consistent method of identifying these patients in the preoperative setting for potential intervention. As seen in our study, there is a significant added value in triaging patients for sarcopenia with PMI. Incorporating PMI into pre-LVAD risk stratification algorithms could help direct high-risk patients toward intensified prehabilitation strategies, potentially improving both early and late outcomes.

### 4.6. Limitations

This study has several limitations that should be considered when interpreting the results. First, the limited sample size, particularly within the sarcopenic cohort, may have contributed to non-significant *p*-values despite clinically meaningful trends and limited the statistical power of our findings. Additionally, as a retrospective study, it is inherently subject to selection bias and potential inaccuracies in the medical records from which data were extracted. The retrospective design also limits our ability to establish causality between sarcopenia and outcomes following LVAD implantation. Another important limitation is the inability to fully control for comorbid conditions, such as chronic kidney disease, diabetes, and atrial fibrillation, which are prevalent in this patient population and may confound the relationship between sarcopenia and postoperative outcomes. Although we attempted to compare these factors between groups, unmeasured or residual confounding could still be present.

### 4.7. Future Directions

Future research should focus on investigating the potential for recovery of the psoas muscle area following LVAD implantation and its correlation with long-term outcomes. Given the association between sarcopenia and poor perioperative results, it would be valuable to explore whether improvements in muscle mass, particularly the psoas muscle, occur as patients stabilize hemodynamically with LVAD support. Such studies could incorporate serial imaging to track changes in PMI over time, coupled with functional measures such as grip strength, 6 min walk test distance, and patient-reported quality-of-life scores. Linking anatomical recovery to functional recovery would help clarify whether muscle mass restoration directly translates to improved independence and reduced morbidity. Furthermore, studying the factors that promote muscle recovery, such as nutritional interventions, physical therapy, and structured cardiac rehabilitation, could identify strategies to optimize pre- and post-implantation management for sarcopenic patients. Ultimately, utilization of PMI in the preoperative setting to stratify patients based on 90-day mortality risk could allow for patient-centered multimodal prehabilitative interventions that may increase the long-term mortality benefits observed in the non-sarcopenic cohort and LVAD recipients at large.

## 5. Conclusions

PMI-defined sarcopenia is associated with significantly worse early postoperative survival in our study. This highlights the critical role of skeletal muscle reserves in surgical recovery and suggests that PMI could serve as a practical tool for risk stratification in LVAD candidates. Moreover, early identification of sarcopenia in LVAD candidates could enable targeted nutritional and rehabilitative interventions to improve perioperative outcomes. Larger prospective studies will be necessary to validate our findings and provide a more comprehensive understanding of the impact of sarcopenia on LVAD outcomes and whether interventions can impact survival in this high-risk population.

## Figures and Tables

**Figure 1 muscles-04-00041-f001:**
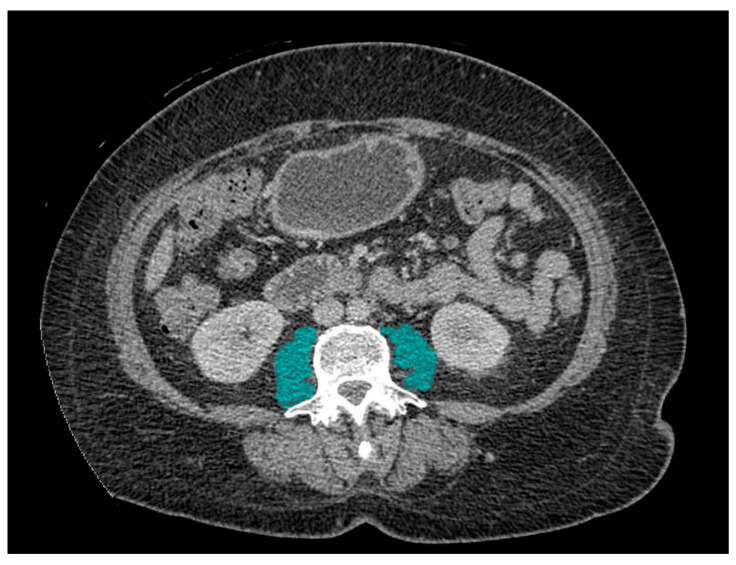
Representative CT imaging of the psoas muscle. Axial CT slice at the level of L3 showing bilateral psoas muscles (blue shading) used to calculate the PMI.

**Figure 2 muscles-04-00041-f002:**
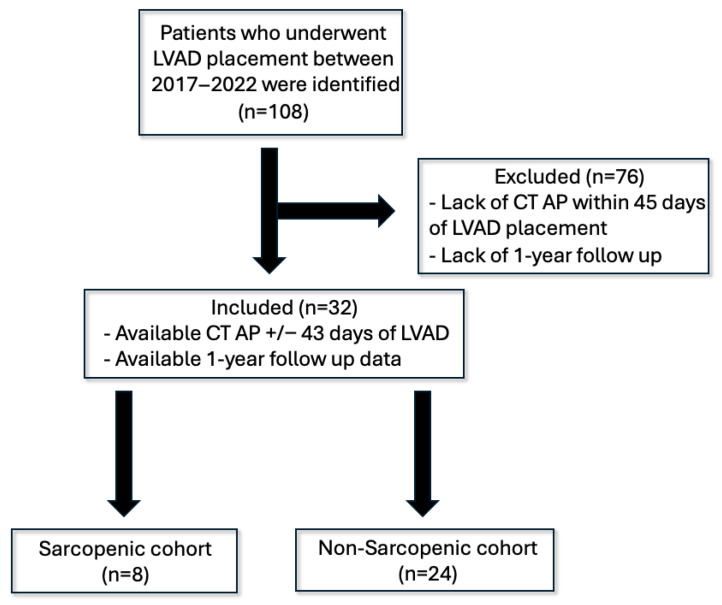
Exclusion criteria and cohort formation. Legend: CT AP: computed tomography abdomen/pelvis. LVAD: left ventricular assist device.

**Figure 3 muscles-04-00041-f003:**
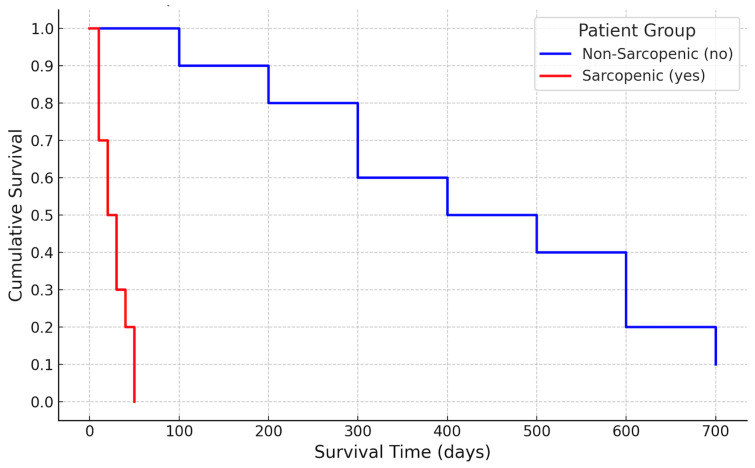
Kaplan–Meier survival curve (visual take-home graphic).

**Table 1 muscles-04-00041-t001:** Demographic information and comorbidities.

	Non-Sarcopenic (24)	Sarcopenic (8)	*p* Value
Gender (Male)	17 (70.8%)	7 (87.5%)	0.346
Ethnicity			0.396
Caucasian	7 (29.2%)	5 (62.5)	
African American	11 (45.8%)	2 (25%)	
Hispanic	5 (20.8%)	1 (12.5)	
Asian	0	0	
Other	1 (4.2%)	0	
Age at Transplant—Average	55.8	60	0.271
Chronic Kidney Disease	18 (75%)	6 (75%)	1
Atrial Fibrillation	5 (20.8%)	2 (25%)	0.805
Myocardial Infarction	5 (20.8%)	3 (37.5%)	0.660
Liver Disease	5 (20.8%)	3 (37.5%)	0.660
Cerebral Vascular Accident	9 (37.5%)	4 (50%)	0.533
Outpatient Milrinone Use	11 (45.8%)	3 (37.5%)	0.681
LVAD as BTT	3 (12.5%)	1 (12.5%)	1.0

LVAD: left ventricular assist device. BTT: bridge to therapy.

**Table 2 muscles-04-00041-t002:** Nutritional and functional physical markers.

	Non-Sarcopenic (24)	Sarcopenic (8)	
BMI average	29.6	30.6	0.353
Albumin average	3.17	3.23	0.403
Prealbumin average	19.26	18.36	0.389
Grip strength average	29.59	28.23	0.796
At-risk/malnourished MNA	16 (66.7%)	5 (62.5%)	0.830

BMI: body mass index. MNA: mini nutritional assessment.

**Table 3 muscles-04-00041-t003:** Mortality, length of stay, and overall mortality.

	Non-Sarcopenic (24)	Sarcopenic (8)	*p*-Value
Overall Mortality	9 (37.5%)	3 (37.5%)	1
90-day Mortality	2 (8.3%)	3 (37.5%)	0.16
1-year Mortality	4 (16.6%)	3 (37.5%)	0.459
Post-op LOS average	23.33	26.43	0.623
OS among Deceased (days)	597	38	0.006

Post-op: postoperative. LOS: length of stay. OS: overall survival.

**Table 4 muscles-04-00041-t004:** Kaplan–Meier survival curve data.

Sarcopenic Status	Mean Estimate (days)	Std. Error	95% CI Lower Bound (Mean)	95% CI Upper Bound (Mean)	Median Estimate	Std. Error (Median)	95% CI Lower Bound (Median)
no	597	171.648	260.57	933.43	415	38.759	339.033
yes	38	14.012	10.537	65.463	25	1.633	21.799
Overall	457.25	146.308	170.486	744.014	374	62.354	251.786

Std. Error: standard error. CI: confidence interval.

## Data Availability

The datasets presented in this article are not readily available because the data are part of an ongoing study. Requests to access the datasets should be directed to Temple University Hospital.

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
