# Peer review of "Psoas Muscle Index Predicts Perioperative Mortality in Patients Undergoing LVAD Implantation"

_muscles, 2025, doi:10.3390/muscles4030041_

Round 1
Reviewer 1 Report
Comments and Suggestions for Authors
This study's statistical analysis is limited by the small number of outcomes. However, it is interesting to note that three of the eight patients with low PMI died early after LVAD implantation. A larger sample size may have revealed significant differences in some of the parameters investigated. As the authors state, further study with a larger sample size is warranted.
This paper comprehensively covers the background, methods, and results, and the discussion appears well-written. However, the following points regarding abbreviations and typos should be noted. Finally, the criteria for sarcopenia used by EWGSOP are now commonly used. I recommend mentioning this in the discussion.
Line 108
For HGS and MNA, spell out the full name first, followed by the abbreviation in parentheses.
Line 131
For CTAP, spell out the full name first, followed by the abbreviation in parentheses.
Line 187
Isn't it a mistake to say "mean OS" instead of "median OS"?
Author Response
Reviewer 1, Comment (Line 108):
For HGS and MNA, spell out the full name first, followed by the abbreviation in parentheses.
Response:
We thank the reviewer for this helpful suggestion. We have revised the Methods section to spell out hand grip strength (HGS) and Mini Nutritional Assessment (MNA) in full at their first mention.
Reviewer 1, Comment (Line 131):
For CTAP, spell out the full name first, followed by the abbreviation in parentheses.
Response:
We appreciate the reviewer’s attention to detail. We have updated the manuscript to spell out computed tomography of the abdomen and pelvis (CTAP) at first mention for clarity.
Reviewer 1, Comment (Line 187):
Isn't it a mistake to say "mean OS" instead of "median OS"?
Response:
Thank you for pointing this out. We agree with the reviewer that median OS is the more accurate descriptor in this context. We have corrected the text accordingly to ensure consistency with the results presented in Table 4 and the Kaplan–Meier analysis.

Reviewer 2 Report
Comments and Suggestions for Authors
Dear Research Team,
First of all, thank you for the opportunity to review your work. It is always an enriching experience and a great opportunity to learn more about these important topics.
Secondly, I found your study very interesting, as CT scans are currently widely used in the assessment of sarcopenia, along with muscle ultrasound, of which I am a strong advocate.
Thirdly, I would like to share some suggestions that you may consider useful to improve the manuscript:
a) It would be advisable to include a section within the methodology discussing the sample size and how it was calculated, in order to assess whether the statistical significance of your results may be related to this aspect.
b) I suggest including a specific subsection within the methodology clearly outlining the inclusion and exclusion criteria. While reading, I had difficulty locating this information.
c) Please consider adding a bibliographic reference for the formula used to calculate the PMI (Psoas Muscle Index).
d) To improve the readability of the manuscript, I recommend structuring both the results and the discussion sections according to the different parts of your study. This would help guide the reader more effectively through the content.
e) Including a dedicated “Conclusions” section would be beneficial.
f) Do you believe the non-significant p-values observed in your results may be due to a limited sample size? Perhaps collaborating with other hospitals could help increase your sample and potentially yield more robust results. Currently, sarcopenia is recognized as a marker of poor prognosis, longer hospital stays, increased costs, mortality, etc., so the lack of differences between sarcopenic and non-sarcopenic patients is striking.
g) It would greatly enhance your work to include iconography—specifically, images of the CT scan showing the measurement markings, along with reference values presented in a graphical format. This would add significant value to your publication.
Once again, congratulations on your research, and best wishes as you continue refining your work.
Best regards,
Author Response
Reviewer 2, Comment (a):
It would be advisable to include a section within the methodology discussing the sample size and how it was calculated, in order to assess whether the statistical significance of your results may be related to this aspect.
Response:
We thank the reviewer for this insightful suggestion. We have clarified in the Methods section that, as this was a retrospective study, the sample size was determined by the number of patients who underwent LVAD implantation with preoperative CT imaging available within 45 days. No a priori power calculation was performed, and we have further noted in the Limitations that the small sample size may have influenced statistical significance.
Reviewer 2, Comment (b):
I suggest including a specific subsection within the methodology clearly outlining the inclusion and exclusion criteria. While reading, I had difficulty locating this information.
Response:
We agree and have revised the Methods to include a dedicated subsection entitled Inclusion and Exclusion Criteria. These criteria are now explicitly described to improve clarity and readability.
Reviewer 2, Comment (c):
Please consider adding a bibliographic reference for the formula used to calculate the PMI (Psoas Muscle Index).
Response:
We appreciate this recommendation. We have added citations supporting the PMI calculation method, specifically referencing Bahat et al. (Clin Nutr, 2021) and Balsam (J Thorac Dis, 2018).
Reviewer 2, Comment (d):
To improve the readability of the manuscript, I recommend structuring both the results and the discussion sections according to the different parts of your study. This would help guide the reader more effectively through the content.
Response:
We thank the reviewer for this helpful suggestion. Both the Results and Discussion sections have been restructured into subsections (e.g., Cohort and Sarcopenia Distribution, Nutrition and Functional Assessments, Clinical Outcomes, Key Findings, Comparison with Existing Literature, Potential Mechanisms, etc.) to guide the reader more clearly through our findings.
Reviewer 2, Comment (e):
Including a dedicated “Conclusions” section would be beneficial.
Response:
We agree and have added a standalone Conclusion section summarizing the principal findings and clinical implications of this study.
Reviewer 2, Comment (f):
Do you believe the non-significant p-values observed in your results may be due to a limited sample size? Perhaps collaborating with other hospitals could help increase your sample and potentially yield more robust results. Currently, sarcopenia is recognized as a marker of poor prognosis, longer hospital stays, increased costs, mortality, etc., so the lack of differences between sarcopenic and non-sarcopenic patients is striking.
Response:
We agree with the reviewer that our limited sample size may have contributed to non-significant p-values despite clinically meaningful trends. This limitation is now explicitly noted in the manuscript. In addition, we acknowledge in Future Directions that multicenter collaborations with larger cohorts will be essential to validate these findings and strengthen statistical power.
Reviewer 2, Comment (g):
It would greatly enhance your work to include iconography—specifically, images of the CT scan showing the measurement markings, along with reference values presented in a graphical format. This would add significant value to your publication.
Response:
We thank the reviewer for this excellent suggestion. We have added a representative axial CT image at the L3 vertebral level as Figure 1, demonstrating bilateral psoas muscle measurement used to calculate the PMI. Reference quartile cutoffs for sarcopenia (4.99 cm²/m² for males, 4.09 cm²/m² for females) are also described in the Methods section. We believe this addition strengthens the clarity and educational value of the manuscript.
Round 2
Reviewer 2 Report
Comments and Suggestions for Authors
I thank the authors for the changes.
no I think it is properly adapted to be published